# The Interaction Between Orientin and the Spike of SARS-CoV-2: An In Silico and Experimental Approach

**DOI:** 10.3390/v18010061

**Published:** 2025-12-31

**Authors:** Gabriel Cavalcante Pacheco, Michele de Sá Ribeiro, Camila Silva de Magalhães, Fabiana Avila Carneiro

**Affiliations:** 1Núcleo Multidisciplinar de Pesquisa em Biologia—NUMPEX-BIO, Campus Duque de Caxias Professor Geraldo Cidade, Universidade Federal do Rio de Janeiro, Rio de Janeiro 25240-005, Brazil; gabriel.cavalcanttep@gmail.com; 2Laboratório de Ultraestrutura Celular Herta Meyer, Centro de Pesquisa em Medicina de Precisão, Instituto de Biofísica Carlos Chagas Filho, Universidade Federal do Rio de Janeiro, Rio de Janeiro 21941-853, Brazil; cheleribeiro5@gmail.com; 3Núcleo Multidisciplinar de Pesquisa em Computação—Numpex-COMP, Campus Duque de Caxias Professor Geraldo Cidade, Universidade Federal do Rio de Janeiro, Rio de Janeiro 25240-005, Brazil; camila@xerem.ufrj.br

**Keywords:** Orientin, SARS-CoV-2, molecular docking, spike protein, antiviral activity

## Abstract

SARS-CoV-2, the causative agent of COVID-19, has led to over seven million deaths worldwide prior to May 2025. Despite widespread vaccination programs, COVID-19 remains a persistent global health challenge, underscoring the urgent need for new therapeutic approaches. Orientin is a flavonoid with reported antiviral activity, though its potential against SARS-CoV-2 remains poorly explored. This study aimed to investigate whether Orientin interacts with the viral Spike protein and impacts viral replication. Molecular docking simulations using DockThor were employed to predict the binding affinity between Orientin and the receptor-binding domain (RBD) of the Spike protein. Fluorescence spectroscopy assays were performed to assess direct interactions between Orientin and the trimeric form of the Spike protein. Additionally, cytotoxicity and viral replication assays were carried out in Vero cells to evaluate Orientin’s antiviral effects. Docking results indicated that Orientin likely binds to key RBD residues involved in ACE2 receptor recognition. Spectroscopic analyses showed a decrease in intrinsic tryptophan fluorescence, suggesting direct interaction. Orientin demonstrated no cytotoxicity in Vero cells and exhibited moderate inhibition of viral replication. These findings suggest that Orientin interacts with critical regions of the Spike protein and may act as a moderate in vitro inhibitor of SARS-CoV-2, warranting further investigation into its therapeutic potential.

## 1. Introduction

Coronaviruses (CoVs) infect a wide range of vertebrate hosts, including humans, primarily due to the genetic plasticity of their RNA genomes. Mutations and recombination events, especially in the gene encoding the Spike (S) protein, enable CoVs to adapt to new hosts and exhibit broad cellular tropism [1,2]. Since the first human coronavirus (HCoV) was identified in 1960, seven additional HCoVs have been discovered, including SARS-CoV-2, which emerged in late 2019 in Wuhan, China, and led to the COVID-19 pandemic declared by the World Health Organization (WHO) in March 2020 [3,4,5,6]. The disease has caused an estimated seven million deaths worldwide prior to May 2025 [7].

SARS-CoV-2 is a betacoronavirus sharing 82% genome identity with SARS-CoV and encodes four structural proteins: Spike (S), Nucleocapsid (N), Envelope (E), and Membrane (M) [8,9]. The Spike protein is a transmembrane glycoprotein located on the viral surface and plays a critical role in mediating entry into host cells [10]. It comprises two functional subunits: S1, responsible for receptor recognition, and S2, which mediates membrane fusion [11]. Within S1 lies the receptor-binding domain (RBD), which contains the receptor-binding motif (RBM), a specific region that directly interacts with the host cell receptor angiotensin-converting enzyme 2 (ACE2) [12]. This binding is essential for viral attachment, entry, and subsequent replication, making the Spike protein, and particularly the RBD and RBM, key targets for therapeutic intervention [11,12,13].

Although vaccines have significantly reduced SARS-CoV-2 transmission and severe outcomes, the emergence of immune-evasive variants such as Omicron and declining immunity over time have necessitated booster doses and updated vaccine formulations [14,15,16,17,18,19]. Nonetheless, limited booster uptake and ongoing viral circulation continue to sustain the risk of new variants that may partially evade immune protection [20,21,22,23]. Thus, the development of antiviral drugs remains essential as a complementary strategy to vaccination.

Orientin (Figure 1) is a water-soluble C-glycoside flavonoid (C_21_H_20_O_11_, 448.37 g/mol) found in plants such as *Ocimum tenuiflorum*, *Trollius chinensis*, and *Passiflora* species [24,25]. It exhibits antioxidant, anti-inflammatory, and antiviral properties, including activity against parainfluenza virus type 3 and dengue virus type 2 [26,27]. However, its potential role against SARS-CoV-2 remains largely unexplored. Existing studies have been limited to computational analyses predicting its interaction with viral proteins, including the Spike protein [28,29,30,31].

To complement the limited experimental data currently available on Orientin’s antiviral potential against SARS-CoV-2, this study evaluates its ability to bind the Spike protein’s receptor-binding domain and inhibit viral replication through integrated in silico and in vitro approaches.

## 2. Materials and Methods

### 2.1. Computational Methodology

#### 2.1.1. Ligand and Protein Preparation

The three-dimensional structure file (SDF format) of Orientin (PubChem CID: 5281675) was downloaded from the PubChem database, which is accessible through the United States National Institutes of Health (NIH) online platform [32]. The flavonoid’s tautomeric states and protonation were predicted at pH 7.4 (with a tolerance of ±1.0) using both the classical and machine learning-based versions of the Epik tool from the Schrödinger platform (Schrödinger Inc. New York, USA) [33,34]. The preparation of the ligand followed the protocol previously described by Guedes and colleagues [35]. The protein structures for the docking experiments were available in the DockThor program (Laboratório Nacional de Computação Científica—LNCC, Rio de Janeiro, Brasil) and prepared using the PROPKA program from the Schrödinger platform [36]. This process included adding hydrogen atoms (pH = 7), removing water molecules, ions, cofactors, and ligands, and converting selenomethionine residues to methionine.

#### 2.1.2. Molecular Docking

The molecular docking experiments were conducted using the DockThor program [37,38]. We utilized the three-dimensional structure of the receptor-binding domain (RBD) of the wild-type SARS-CoV-2 Spike protein. The structural files of the domains were available in the “COVID-19 resources” section of DockThor’s “Docking” tool. The RBD structure used was determined in two ways: complexed with angiotensin-converting enzyme 2 (ACE2) (PDB: 6M0J) [39] or complexed with neutralizing antibodies that inhibit SARS-CoV-2 binding to ACE2 (PDB: 7BZ5) [40].

Three docking experiments were performed using these structures. In the first experiment, we used the “PDBcode_6m0j-noACE2” file from DockThor, corresponding to PDB 6M0J without the ACE2 structure. We used the “PDBcode_7BZ5” file for the second experiment, using only the RBD structure. The glutamine 493 and asparagine 501 (Asn501) amino acid residues, which are important for protein–protein interactions, assume different positions in both structures. Therefore, using these structures allows an analysis of how Orientin interacts with the RBD in different conformations of these residues. In the third experiment, we used the complete PDB 6M0J structure (DockThor file: PDBcode_6m0J-ACE2), which includes the RBD-ACE2 complex.

Docking was performed without the inclusion of cofactors. The energy grid configuration for the binding site was built by centering the grid on the Gln493 residue (Cα), with spatial coordinates X = −39.9, Y = 31.0, and Z = 7.5. The box size was set to 22 Å in each axis, with a discretization of 0.25. These parameters were suggested by the program for these targets. To define the precision of the search algorithm, we used DockThor’s default parameters, which included 24 program runs, 1,000,000 function evaluations per docking, and a population of 750 individuals. All images of protein–ligand interactions were generated using the Discovery Studio (Version 2025, Dassault Systèmes, Johnston, USA) and Pymol (Version 3.0, Schrödinger Inc. New York, USA) programs [41,42].

### 2.2. Experimental Methodology

#### 2.2.1. Fluorescence Quenching

Intrinsic fluorescence analysis assays were performed to evaluate the interaction between Orientin and the 18 tryptophan residues of the SARS-CoV-2 Spike protein [43]. For these assays, we used the prefusion conformation of the Spike protein trimer (batch: M180221) [44], kindly provided by the Laboratório de Genômica Molecular (LABGENEST) at the Universidade Federal do Rio de Janeiro, and 97% ultrapure Orientin was purchased from Sigma-Aldrich (batch: MKCN755. Merck, Darmstadt, Germany). Fluorescence experiments were conducted using a Cary Eclipse/Varian fluorescence spectrophotometer. A total of 100 µL (118 µg) of Spike protein was added to 500 µL of PBS, followed by increasing concentrations of Orientin at 5, 10, 20, 50, and 100 µM. The molecules were excited at 280 nm, and fluorescence emission spectra were recorded between 270 and 420 nm. The intrinsic fluorescence emission data of the protein’s tryptophan residues were analyzed using the Stern–Volmer equation [45]:(1)$$\frac{I_{0}}{I}=\left(1+K_{q}\tau 0\left[Q\right]\right)=\left(1+K_{sv}\left[Q\right]\right)$$
where I_0_ represents the fluorescence emission intensity of the Spike protein in the absence of Orientin, I is the fluorescence emission intensity after the addition of Orientin, K_q_ is the biomolecular quenching constant, τ0 is the fluorophore lifetime in the absence of a quencher (for tryptophan, a value of 10^−8^ s) [45], Q represents the concentration of Orientin, and K_sv_ is the Stern–Volmer constant.

#### 2.2.2. Cell, Compound, and Virus

Vero cells were purchased from ATCC. Orientin was purchased from Sigma-Aldrich (St. Louis, MO, USA). SARS-CoV-2 was kindly given by Dr. Amilcar Tanuri (Instituto de Biologia—UFRJ).

#### 2.2.3. Cell Culture

A fibroblast cell line derived from the kidney of the African green monkey Cercopithecus aethiops (Vero cells) was used to investigate virus replication. Vero cells were cultured in complete medium consisting of DMEM (Dulbecco’s Modified Eagle’s Medium—Gibco) containing 4.5 g/L glucose, 2 g/L sodium bicarbonate, 2 mM L-glutamine, 100 U/mL penicillin (Gibco), 100 μg/mL streptomycin (Gibco), and 2.5 μg/mL amphotericin B, supplemented with 2% (cell maintenance medium) or 10% (growth medium) inactivated fetal bovine serum (FBS; Merck, Darmstadt, Germany). Cultures were maintained at 37 °C in a 5% CO_2_ atmosphere.

#### 2.2.4. Cell Viability Assay

Vero cells were seeded at a density of 0.1 × 10^6^ cells per well in 96-well plates and used at approximately 80% confluence. The cells were treated with Orientin at concentrations of 5, 10, 50, and 100 μM for 48 h. After the incubation period, cell viability was assessed using the MTS/PMS protocol (Invitrogen; Thermo Fisher Scientific Inc., Waltham, MA, USA). The culture supernatant was removed, and 100 μL of electroporation medium was added to each well. Next, 20 μL of MTS solution containing 1 μL of PMS was added to each well. The plate was covered with aluminum foil and incubated at 37 °C for 4 h. Absorbance was measured at 490 nm (endpoint), according to the manufacturer’s instructions.

#### 2.2.5. Antiviral Activity

The antiviral activity of Orientin was assessed using the TCID_50_ (infectious dose in tissue culture 50%) assay in 96-well plates, with 10,000 cells seeded per well. The cells were infected with serial dilutions of SARS-CoV-2 and incubated for 1 h at 37 °C in a 5% CO_2_ atmosphere. After the adsorption period, the viral inoculum was removed, and the cells were treated with Orientin at concentrations of 5, 10, and 20 µM. Following 48 h of incubation, the viral titer was determined in both the presence and absence of Orientin. The percentage of viral cytopathic effect inhibition was calculated by comparing the viral titers of the untreated infected control group with those of the treated groups, according to the method described by Reed and Muench [46].

#### 2.2.6. Virucidal Activity

The viral suspension was made in the serial dilution system (101–1010) in complete medium without FBS, in the presence or absence (control) of Orientin at 5, 10, and 20 µM, and incubated at 4 °C for 1 h. After the designated period, Vero cells were distributed in 96-well plates (1 × 10^4^ cells/mL) and infected with the viral suspensions, which were incubated for 1 h at 37 °C. At 48 h.p.i., the viral titer was determined using the TCID_50_ method, corresponding to a 50% cytopathic effect in cells.

## 3. Results

### 3.1. Molecular Docking Reveals That Orientin Interacts with RBM

The molecular docking results are summarized in Table 1, which ranks the predicted binding free energies for each lowest-energy pose of Orientin interacting with the RBD, ordered by total energy. Regarding the predicted binding energy in the first molecular docking, where we used the “PDBcode_6m0j-noACE2” file, the best poses were found in runs 2, 7, and 23, with predicted binding free energies of −6.75 kcal/mol, −6.95 kcal/mol, and −6.56 kcal/mol, respectively.

Figure 2 shows the binding pose of Orientin to the RBM, based on the RBD structure from PDB 6M0J (without ACE2), corresponding to binding mode 2 reported in Table 1. The flavonoid forms three hydrogen bonds with the amino acid residues of the RBM: Glu484, Leu492, and Ser494. Additionally, van der Waals interactions between Orientin and the RBM residues Tyr449, Phe456, Leu455, Tyr489, Phe490, and Gln493 also contribute to the intermolecular interactions between the protein and the ligand.

In the second molecular docking, we used the “PDBcode_7BZ5” protein file, which corresponds to the Spike protein originally complexed with antibodies, as provided by the DockThor platform. The binding modes of Orientin with the lowest total energy, distinct by at least 2 Å, were identified in execution runs 15, 6, and 23 (Table 1), with predicted binding free energies of −7.00 kcal/mol, −6.51 kcal/mol, and −6.78 kcal/mol, respectively. The most favorable binding mode (lowest total energy) corresponds to the lowest predicted binding free energy value for Orientin interacting with the RBM (pose number 15). In this binding mode, Orientin forms three hydrogen bonds with the amino acid residues Gln493, Glu406, and Lys417 (Figure 3). The Lys417 residue interacts with Orientin via a non-covalent pi-alkyl bond. Another non-covalent cation-pi interaction occurs between this binding mode of the flavonoid and the Arg403 residue. Additionally, van der Waals interactions are established with Ile418, Tyr495, Tyr505, Tyr453, Ser494, and Leu455. 

Beyond the predicted binding free energy values and total energies, the interactions between Orientin and the RBD showed notable differences in the best binding modes found in each docking with the different target structures (6M0J and 7BZ5). However, in both molecular dockings, Orientin consistently interacted with Leu455 via van der Waals interactions. In the first molecular docking (6M0J pdb file), the lowest-energy pose from run 2 showed that Orientin formed a hydrogen bond with Ser494 and engaged in van der Waals interactions with Gln493. Conversely, in the second molecular docking (7BZ5 pdb file), the lowest-energy pose identified in run number 15 revealed that Orientin interacted with Ser494 via van der Waals forces and formed a hydrogen bond with Gln493. As shown in Figure 2 and Figure 3, Orientin may adopt distinct poses when interacting with the RBD. This difference is likely influenced by the conformational variations of Gln493 and Asn501 in the RBD’s three-dimensional structure, as shown in the two structure files.

In the molecular docking using the full structure of RBD-ACE2 (Dockthor file: PDBcode_6M0J-ACE2), Orientin was found to interact with several key residues within the receptor-binding motif (RBM) of the Spike protein, including Leu452, Leu492, Glu484, Phe490, Ser494, Gln493, and Tyr449. In addition, binding interactions were also observed with critical residues of the angiotensin-converting enzyme 2 (ACE2), specifically Asp38, Glu35, Lys31, and Leu39 (Figure 4).

Furthermore, the predicted binding free energy of the best binding mode found was −7.44 kcal/mol (Table 1). In this docking, Orientin’s most favorable binding mode exhibited the lowest total energy and the lowest predicted binding affinity in relation to all other dockings performed in this work. This suggests that Orientin may interact more favorably with ACE2, which could contribute to the lower energy values observed.

Orientin’s pose when interacting with the RBM of Spike protein of SARS-CoV-2 at run 15 of molecular docking using 7BZ5 pdb file shows that the ligand remains within the RBD-ACE2 interaction site. However, the molecule occupies a more distant position than the best (lowest-energy) poses that Orientin adopts when interacting with RBM in PDB 6M0J, with or without ACE2 (Appendix A). This is also evidenced by the fact that the best pose of Orientin (run 15 of the docking using the pdb file 7BZ5) interacts with a different set of amino acid residues compared to those identified in the interaction between the best poses of Orientin generated in the dockings using the file 6M0J (with and without ACE2) (Appendix A).

Furthermore, the hydrogen bonds formed between Glu484 and Ser494 of the Spike RBD and the best poses of Orientin from molecular dockings 1 and 3 (in which the pdb file 6MOJ with and without ACE2 were used) are similar, although the atoms involved in these interactions, both in the ligand and in the receptor, belong to different functional groups (Appendix A).

### 3.2. The Decrease in the Intrinsic Fluorescence of the Spike Protein Suggests an Interaction with Orientin

The fluorescence emission spectra of tryptophan residues in the SARS-CoV-2 Spike protein were recorded in the presence of increasing concentrations of Orientin (5, 10, 20, 50, and 100 μM) (Figure 5). A dose-dependent decrease in fluorescence intensity was observed, indicating a quenching effect of Orientin on the intrinsic fluorescence of the Spike protein (Figure 5A). However, no significant shift in emission wavelength was detected across the tested concentrations, suggesting the absence of major conformational rearrangements in the protein’s tertiary structure.

Shifts in fluorescence emission wavelength are widely used to probe conformational changes in proteins following ligand binding or changes in physicochemical conditions [47]. Typically, a red shift (toward longer wavelengths) indicates greater exposure of tryptophan residues to a polar (hydrophilic) environment, while a blue shift (toward shorter wavelengths) suggests internalization of these residues into a more hydrophobic core [47]. The near-constant emission maximum observed here (0–1 nm red shift) suggests that Orientin does not cause notable perturbations to the structural environment of tryptophan residues within the Spike protein.

Despite the absence of wavelength shifts, the marked reduction in fluorescence intensity upon addition of Orientin points to a quenching interaction. Fluorescence quenching can occur via dynamic (collisional) or static (complex formation) mechanisms [48]. Dynamic quenching results from non-radiative energy transfer during collisions between the fluorophore in its excited state and the quencher, whereas static quenching arises from ground-state complex formation that prevents excitation and subsequent emission.

Analysis of the Stern–Volmer plot (Figure 5B) reveals that fluorescence quenching increases with increasing Orientin concentration. The slight upward curvature observed suggests the presence of both dynamic and static components. However, this curvature was subtle and insufficient to distinguish the predominant mechanism, especially given the structural complexity of the Spike glycoprotein, which contains 18 tryptophan residues distributed across its domains. Importantly, the upward deviation in the plot—curving toward the *y*-axis—is indicative of a mixed quenching mechanism [49].

To further characterize the quenching behavior, we calculated the bimolecular quenching rate constant (K_q_) using the Stern–Volmer constant (K_sv_) derived from the linear fit of the experimental data. Applying the equation K_q_ = K_sv_/τ0, where τ0 represents the fluorescence lifetime of tryptophan (~10^−8^ s), we obtained a K_q_ value of 3.53 × 10^12^ M^−1^s^−1^. This magnitude is consistent with a static quenching mechanism and supports the hypothesis that Orientin quenches tryptophan fluorescence by forming a complex with Spike protein.

Based on this assumption, we determined the association constant (K_a_), considering that the slope obtained from the Stern–Volmer equation is numerically equivalent to K_a_ (K_a_ = K_sv_), which is justified by the Law of Mass Action under the static model. After correcting the mean value of the Stern–Volmer Constant to M^−1^, the experimental results revealed a mean dissociation constant (K_d_) of 32.72 ± 6.22 µM.

Notably, the high R^2^ value (>0.90) obtained from our linear regression analysis indicates low experimental variability, thereby supporting the reproducibility and reliability of the quenching data. The interaction between Orientin and the Spike protein is substantiated by both the progressive fluorescence quenching of tryptophan residues and the calculated K_q_ value. Control assays using PBS and Orientin alone (Appendix A) confirmed that neither buffer component nor Orientin itself emits fluorescence within the tryptophan emission range, ruling out false-positive signals and affirming the specificity of the quenching interaction.

### 3.3. Orientin Reduces SARS-CoV-2 Replication in Vero Cells

Cell viability and viral replication assays were performed using Vero cells to determine if Orientin has antiviral effects by decreasing the number of viral particles produced during replication. Orientin was evaluated for its cytotoxicity against Vero cells using the MTS assay. Figure 6 shows that Orientin exhibited no significant cytotoxicity in Vero cells after 48h of incubation at increasing concentrations.

The antiviral activity of Orientin was assessed by EC_50_ values and virucidal effects against SARS-CoV-2 using the TCID_50_ assay (Table 2 and Table 3). The untreated virus reached a titer of 7.05 × 10^6^ TCDI_50_/mL, serving as the control.

Treatment with Orientin at 20 µM reduced viral replication to 7.92 × 10^4^ pfu/mL, corresponding to a ~2.0 log_10_ reduction compared to control. The virucidal assay at this concentration demonstrated an even stronger effect, reducing viral titers to 2.10 × 10^4^ pfu/mL, equivalent to a 2.54 log_10_ reduction (Table 2 and Table 3). At 10 µM, Orientin inhibited replication to 1.00 × 10^5^ pfu/mL (1.85 log_10_ reduction), while virucidal activity yielded 2.34 × 10^5^ pfu/mL (~1.6 log_10_ reduction) (Table 2 and Table 3). At the lowest concentration tested (5 µM), replication was reduced to 2.09 × 10^5^ pfu/mL, representing ~1.5 log_10_ reduction, whereas virucidal activity decreased titers to 1.10 × 10^5^ pfu/mL, equivalent to ~1.8 log_10_ reduction (Table 2 and Table 3).

Overall, Orientin displayed dose-dependent inhibitory effects in the EC_50_ assay, reaching nearly 2 log_10_ reduction at the highest concentration tested. Importantly, virucidal assays confirmed their ability to directly inactivate viral particles, with the most pronounced effect observed at 20 µM. These findings demonstrate that Orientin exerts both replication-inhibitory and virucidal effects against SARS-CoV-2 in vitro.

## 4. Discussion

As demonstrated by our molecular docking results, Orientin exhibited a predicted binding free energy of approximately −7 kcal/mol. Furthermore, several studies indicate that the amino acid residues Lys417, Gly446, Tyr449, Tyr453, Leu445, Phe456, Gln474, Ala475, Gly476, Ser477, Glu484, Phe486, Asn487, Tyr489, Gln493, Gly496, Gln498, Thr500, Asn501, Gly502, and Tyr505 within the receptor-binding motif (RBM) of the wild-type SARS-CoV-2 Spike protein interact via hydrogen bonding with various residues of ACE2 [50,51]. Thus, considering that the RBM comprises amino acid residues 437–508, the best binding poses of Orientin predicted in the docking experiments conducted in this study suggest that the phytocompound interacts with some of these amino acids that constitute the receptor-binding motif of the SARS-CoV-2 Spike protein.

Several studies have investigated how natural compounds interact with the SARS-CoV-2 Spike protein to identify molecules with potential for antiviral development. Bhowmik et al. demonstrated, through in silico studies, that the natural compounds Orientin, Caffeic Acid, Isobavachalcone, Lycorine, Ellagic Acid, and Galangin interact with the SARS-CoV-2 Spike protein, exhibiting binding free energies of −6.2 kcal/mol, −5.1 kcal/mol, −6.0 kcal/mol, −5.7 kcal/mol, −6.0 kcal/mol, and −5.8 kcal/mol, respectively [28]. Based on these values, they concluded that Orientin has the highest affinity for the Spike protein among the other compounds. Similarly, Lakshmi determined that Orientin interacts with the Spike protein with a computationally predicted binding affinity [29].

The predicted binding free energy values for Orientin found in the molecular dockings conducted in this study ranged from −6.75 kcal/mol to −7.4 kcal/mol. These values suggest a moderate binding free energy between the molecule and the target [35]. In this context, the findings of this study support the results reported by Bhowmik and Lakshmi. Therefore, since the molecular docking results indicate that Orientin exhibits affinity for the RBM of the Spike protein, we hypothesize that, if the phytocompound also interacts with the glycoprotein in vitro, it may attenuate the entry of SARS-CoV-2 into cells.

The aromatic amino acids tryptophan (Trp), tyrosine (Tyr), and, to a lesser extent, phenylalanine (Phe) absorb light at a wavelength of 280 nm and emit intrinsic fluorescence at longer wavelengths [52]. Therefore, proteins containing these amino acids in their structures can be studied using fluorescence spectroscopy, which allows evaluation of the conformational changes these macromolecules undergo and their interactions with ligands. Thus, since the intrinsic fluorescence emission spectra of the tryptophan in the Spike protein exhibit only a slight red shift (0–1 nm), we suggest that Orientin does not induce significant conformational changes in the protein’s tertiary structure.

Additionally, this minor shift can be explained by the fact that the Spike protein used in the fluorescence experiments stabilized in a pre-fusion conformation, preventing it from undergoing the conformational changes required to shift the fluorescence spectra. Therefore, we propose that future experiments be conducted with the protein without stabilization in a specific conformation to determine whether Orientin can induce structural alterations in the Spike protein and reduce its intrinsic fluorescence.

The value of K_q_ = 3.5 × 10^12^ M^−1^s^−1^ obtained in our results indicates that Orientin likely reduces the intrinsic fluorescence of the Spike protein tryptophan through static quenching in a moderate manner. This inference is because the biomolecular quenching constant determined in our experiments is higher than K_q_ = 2.0 × 10^10^ M^−1^s^−1^, which is the maximum collisional quenching constant proposed by Ware [53].

Several studies have evaluated the efficiency of various compounds in quenching the intrinsic fluorescence of human and bovine serum albumins [54,55,56]. These studies demonstrated that the K_q_ values for the tested compounds reached the order of 10^12^ M^−1^s^−1^ and that temperature influenced the value of this constant. The authors observed an inverse relationship between temperature and K_q_, with higher temperatures leading to lower quenching constants. Furthermore, it was determined that since the K_q_ values exceeded 2 × 10^10^ M^−1^s^−1^, the fluorescence quenching was associated with a static quenching effect [54,55,56].

Recently, Xu et al. investigated the non-covalent interaction between seed globulin and phenolic compounds at different temperatures [57]. The K_q_ values were also in the order of 10^12^ M^−1^ s^−1^. However, although temperature was not inversely proportional to the K_q_ value, the authors reported that the fluorescence decay resulted from static quenching, as K_q_ exceeded 2.0 × 10^12^ M^−1^s^−1^, the maximum diffusion-limited quenching constant defined by Ware.

The bimolecular quenching constant values reported in the cited studies are slightly higher than the K_q_ value obtained in this work. Therefore, the interaction efficiency between Orientin and the Spike protein is moderate. However, it is important to note that the studies mentioned employed different proteins and molecules than those used in the present study and evaluated the fluorescence decay as a function of temperature. According to Lakowicz, considering only the linearity of the Stern–Volmer plot is not the most accurate approach to defining the type of quenching responsible for fluorescence emission reduction [49]. Varying the system’s temperature is crucial for distinguishing the underlying quenching mechanism.

The antiviral activity of Orientin observed in this study indicated an inhibitory effect against SARS-CoV-2 in vitro. In the replication assay, orientin reduced viral titers by 1.5 log_10_ at 5 µM and up to 2 log_10_ at 20 µM, indicating a dose-dependent response. In the virucidal assay, direct exposure of viral particles to Orientin resulted in a ~1.8 log_10_ reduction at 5 μM and a maximum reduction of ~2.5 log_10_ at 20 μM. The observed dose dependency is encouraging and consistent with previous reports on the antiviral activity of flavonoids. Quercetin and Baicalein, for example, have demonstrated inhibitory effects against SARS-CoV-2 and other RNA viruses, often with EC_50_ values in the low micromolar range [58,59]. Luteolin, a structurally related flavone, has been shown to block the interaction between the spike protein and the ACE2 receptor, suggesting that Orientin may also interfere with viral entry.

To better contextualize the antiviral profile of Orientin, we compared our results with literature-reported data for three other ligands with known anti-entry activity: cepharanthine and two flavonoids selected as reference compounds, quercetin and myricetin. Cepharanthine is a clinically used bisbenzylisoquinoline alkaloid that has been repurposed as a SARS-CoV-2 entry inhibitor [60,61]. In spike-mediated viral entry assays, cepharanthine displays potent antiviral activity with a literature-reported IC_50_ of 1.30 ± 0.18 µM [60], which is consistent with a high-affinity entry-directed antiviral agent. In contrast, Orientin exhibits an experimental K_d_ of 32.72 ± 6.22 µM, which falls within the low-micromolar range reported for spike/entry-targeting flavonoids. The two reference flavonoids, quercetin and myricetin, exhibit antiviral activity with reported literature IC_50_ values of 17.00 ± 3.42 µM and 10.27 ± 2.32 µM, respectively [62].

Although IC_50_ values from cell-based entry assays and K_d_ values from binding experiments are not directly equivalent, their joint consideration indicates that Orientin behaves as a moderate spike-binding ligand, with an affinity in the same micromolar range as other spike-targeting flavonoids and clearly weaker, yet within the same order of magnitude, than the clinically validated entry inhibitor cepharanthine. This interpretation is also consistent with our docking results, which predict a moderate binding affinity for Orientin (approximately −7 kcal/mol). Scores between approximately −6.8 and −8.2 kcal/mol are indicative of moderate micromolar affinities, whereas scores lower than −8.2 kcal/mol are consistent with submicromolar affinities.

The structural similarity between Orientin and these flavonoids suggests potential for further optimization. Future studies should explore its mechanism of action, possible targets in the viral life cycle, and combinatorial potential with other antiviral agents. Additionally, evaluation in physiologically relevant cell types and in vivo models will be essential to clarify its translational relevance.

In summary, the data from this study suggest that the Spike protein interacts with Orientin, prompting us to evaluate whether this interaction affects SARS-CoV-2 replication in host cells.

## 5. Conclusions

In conclusion, our findings demonstrate that Orientin interacts with key residues of the SARS-CoV-2 Spike receptor-binding motif with consistent affinity, as supported by molecular docking analyses. Intrinsic fluorescence quenching results showed a moderate, static interaction between Orientin and the Spike protein trimer. These biophysical observations are in agreement with the antiviral and virucidal effects observed in vitro, where Orientin reduced SARS-CoV-2 titers in a dose-dependent manner. Together, these results suggest that Orientin may influence the early stages of SARS-CoV-2 entry or particle stability, underscoring its relevance as a natural compound with antiviral properties.

## Figures and Tables

**Figure 1 viruses-18-00061-f001:**
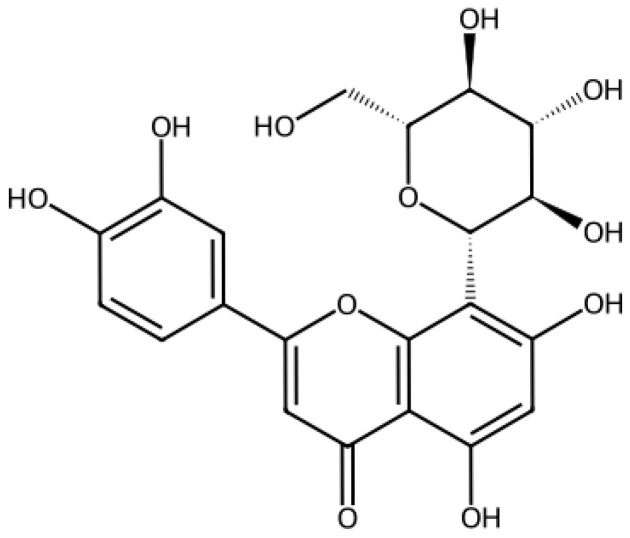
Molecular structure of Orientin.

**Figure 2 viruses-18-00061-f002:**
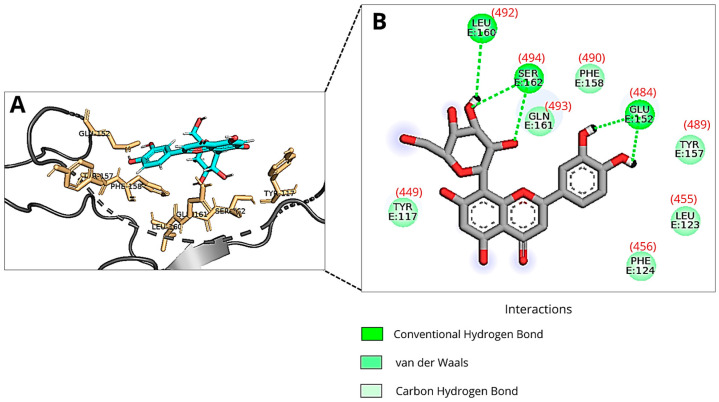
An illustration of Orientin’s pose when interacting with RBM of Spike protein of SARS-CoV-2 at run 2 of molecular docking using 6M0J pdb file (without ACE2 structure): (**A**) Three-dimensional structure of the RBD of the Spike protein (gray) complexed with Orientin (cyan) and the key amino acid residues (orange sticks) that interact directly with the flavonoid; (**B**) The amino acid residues involved in the interaction are represented as green spheres. The green dashed lines indicate hydrogen bonds, and the van der Waals interactions are marked in light green without lines. The amino acid residue numbers inside the circles described in the image correspond to the numbering system adopted in the DockThor protein files, and the numbers in parentheses correspond to the residues 449, 455, 456, 484, 489, 490, 492, 493, and 494 of PDB ID: 6M0J. Images were generated using Discovery Studio and PyMOL.

**Figure 3 viruses-18-00061-f003:**
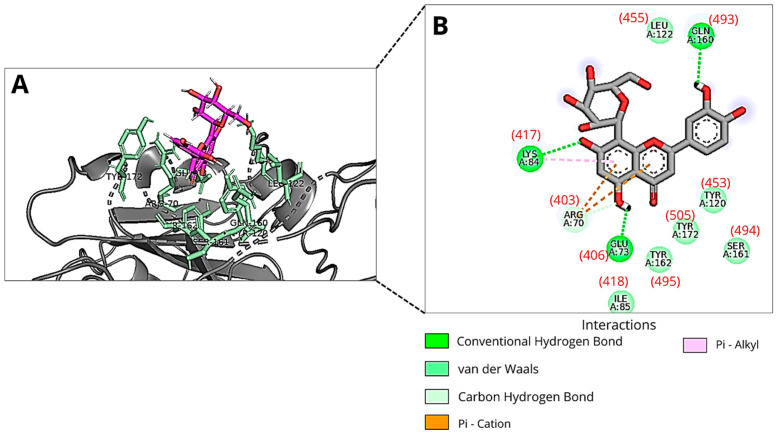
An illustration of Orientin’s pose when interacting with RBM of Spike protein of SARS-CoV-2 at run 15 of molecular docking using 7BZ5 pdb file: (**A**) Three-dimensional structure of the RBD of the Spike protein (gray) complexed with Orientin (magenta) and the key amino acid residues (green sticks) that interact directly with the flavonoid; (**B**) The amino acid residues involved in the interaction are represented as green spheres. The green, orange, and pink dashed lines indicate hydrogen bonds, Pi-Cation and Pi-Alkyl interactions, respectively. The van der Waals interactions are marked in light green without lines. The amino acid residue numbers inside the circles described in the image correspond to the files available in the DockThor program, and the numbers in parentheses correspond to the residues 403, 406, 417, 418, 453, 455, 493, 494, 495, and 505 of PDB ID: 7BZ5. Images were generated using Discovery Studio and PyMOL.

**Figure 4 viruses-18-00061-f004:**
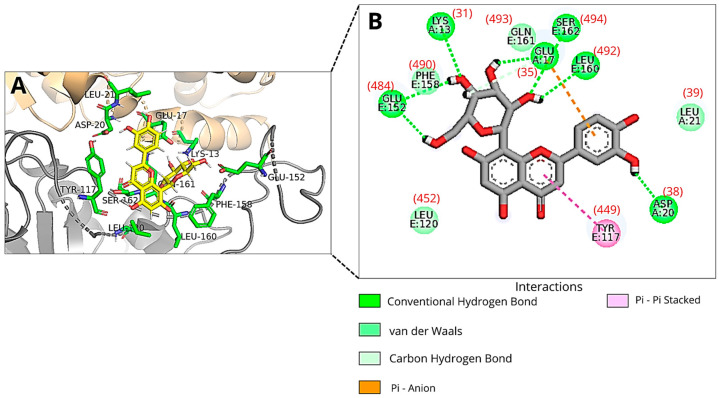
An illustration of Orientin’s pose when interacting with RBM of Spike protein of SARS-CoV-2 and ACE2 at run 4 of molecular docking using the 6M0J pdb file with ACE2: (**A**) Three-dimensional structure of the RBD of the Spike protein (gray) and ACE2 (orange) complexed with Orientin (yellow) and the key amino acid residues (green sticks) that interact directly with the flavonoid; (**B**) The amino acid residues involved in the interaction are represented as green spheres. The green, orange, and pink dashed lines indicate hydrogen bonds, Pi-Anion and Pi-Pi Stacked interactions, respectively. The van der Waals interactions are marked in light green without lines. The amino acid residue numbers inside the circles described in the image correspond to the files available in the DockThor program, and the numbers in parentheses correspond to the residues 31, 35, 38, and 39 (ACE2) and 449, 452, 484, 490, 492, 493, and 494 (RBD) of PDB ID 6M0J. Images were generated using Discovery Studio and PyMOL.

**Figure 5 viruses-18-00061-f005:**
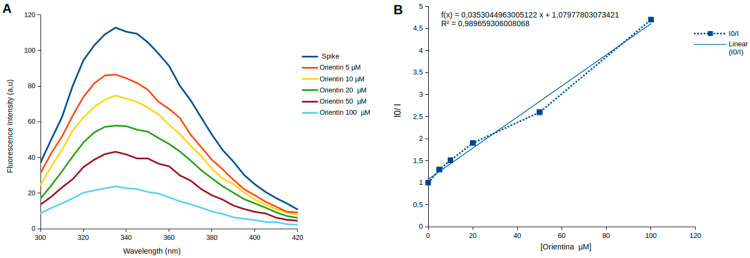
Intrinsic fluorescence spectra of Spike protein titrated with increasing concentrations of Orientin and Stern–Volmer graphics. (**A**) Fluorescence emission spectra of the Spike protein in the absence and presence of increasing concentrations of Orientin (5–100 µM). The dark blue line represents the fluorescence emission spectra of the Spike protein without Orientin. The orange, yellow, green, dark red, and light blue lines represent the Spike protein in the presence of 5, 10, 20, 50, and 100 µM of Orientin, respectively. (**B**) Stern–Volmer plot for the fluorescence quenching of Spike protein by increasing the concentration of Orientin: The graphic shows a correlation between the decrease in the intrinsic fluorescence emission of tryptophan in glycoprotein S and the increase in the concentration of Orientin in the medium. R^2^ = 0.98 indicates low experimental variability.

**Figure 6 viruses-18-00061-f006:**
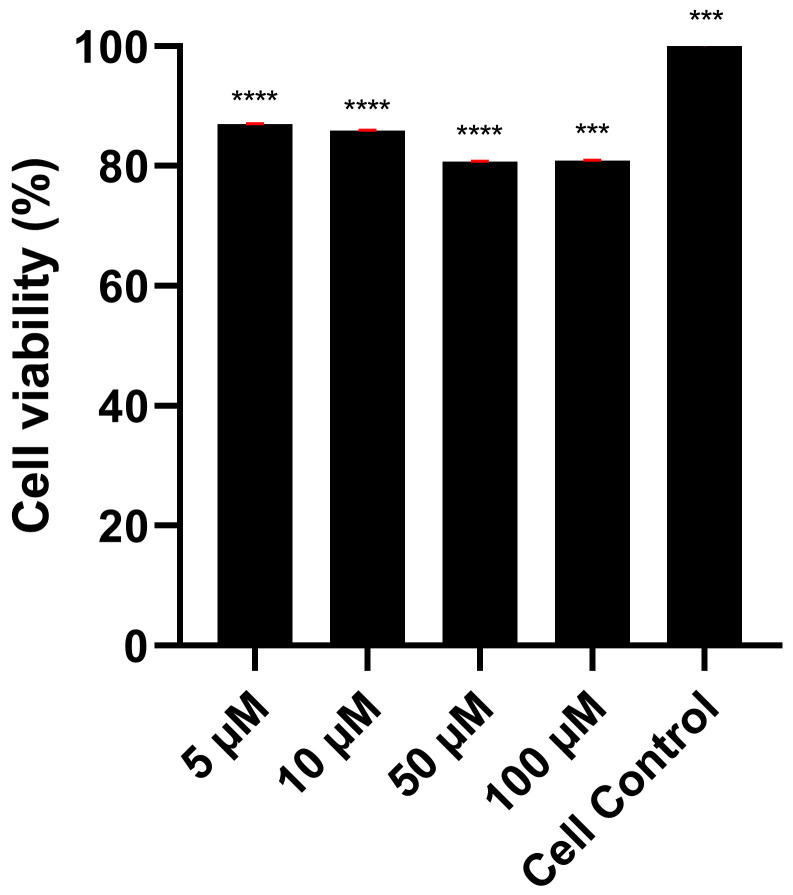
Cytotoxic effect of orientin in Vero cells. Vero cells were treated with increasing concentrations of Orientin for 48 h, and cytotoxic effects were evaluated by MTS assay. All values are the average of nine breeding wells (*** *p* < 0.001, **** *p* < 0.0001).

**Table 1 viruses-18-00061-t001:** Molecular docking results.

PDB File	Binding Mode Number	Binding Free Energy (kcal/mol)	Total Energy (kcal/mol)	van der Waals Energy (vdW) (kcal/mol)	Electrostatic Energy (kcal/mol)
	2	−6.75	53.42	−0.92	−40.23
6M0J (without ACE2)	7	−6.95	53.71	−0.36	−40.85
	23	−6.56	54.22	6.74	−44.47
	15	−7.00	48.60	−0.70	−42.96
7BZ5	6	−6.51	51.43	3.34	−47.79
	23	−6.78	51.64	0.75	−43.30
	4	−7.44	46.46	−9.85	−42.00
6M0J (with ACE2)	3	−6.86	47.75	0.44	−51.02
	15	−6.87	51.87	−6.29	−36.00

**Table 2 viruses-18-00061-t002:** Antiviral activity of Orientin against SARS-CoV-2 in Vero Cells. All values are the average of nine breeding wells (*p* ˂ 0.005).

Treatment Condition	EC_50_ (TCID_50_/mL)	Log_10_ Reduction
Infected Control (no drug)	7.05 × 10^6^ ± 0.025	—
Orientin 5 µM	2.09 × 10^5^ ± 0.06	~1.5 log
Orientin 10 µM	1.00 × 10^5^ ± 0.07	~1.8 log
Orientin 20 µM	7.92 × 10^4^ ± 0.011	~2.0 log

**Table 3 viruses-18-00061-t003:** Virucidal activity of Orientin against SARS-CoV-2 in Vero Cells. All values are the average of nine breeding wells (*p* ˂ 0.005).

Treatment Condition	Virucidal (TCID_50_/mL)	Log_10_ Reduction
Infected Control (no drug)	7.05 × 10^6^ ± 0.08	—
Orientin 5 µM	1.10 × 10^5^ ± 0.013	~1.8 log
Orientin 10 µM	2.34 × 10^5^ ± 0.015	~1.6 log
Orientin 20 µM	2.10 × 10^4^ ± 0.013	~2.5 log

## Data Availability

The original contributions presented in this study are included in the article/Appendix A. Further inquiries can be directed to the corresponding author(s).

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
