# Peer review of "The Interaction Between Orientin and the Spike of SARS-CoV-2: An In Silico and Experimental Approach"

_viruses, 2025, doi:10.3390/v18010061_

Round 1
Reviewer 1 Report
Comments and Suggestions for Authors
This study systematically demonstrates for the first time, through a combined approach of computer simulations and in vitro experiments, that the natural flavonoid compound Orientin interacts with the SARS-CoV-2 spike protein and exhibits moderate antiviral and virus-inactivating activity. It provided important scientific evidence for developing Orientin into a potential anti-COVID-19 drug or lead compound. This study went beyond computational prediction to directly confirm the physical interaction between Orientin and the spike protein trimer through fluorescence spectroscopy experiments, and validated its biological function using cellular models. This multidisciplinary research paradigm enhances the reliability of the conclusions. The study features a rigorous design, substantial experimental data, and significant implications. The paper is worthy of publication, but several revisions are required prior to acceptance.
- The text contains a few instances of non-standard language usage, such as spelling errors like “temp29erature” (P9, line 319).
- We recommend comprehensive language refinement and proofreading throughout the text to ensure consistent terminology. For example, “TCID50, EC50” should be correctly written as “TCID50, EC50” (P4, P12).
- Lack of statistical significance analysis: Neither Figure 5 (cell viability) nor the viral titer results (Tables 2 and 3) provide any statistical significance tests (e.g., p-values). Please supplement the description of the number of experimental replicates and perform comparative statistical analysis on the data to confirm whether the observed effects are statistically significant.
- Lack of statistical significance analysis: Neither Figure 5 (cell viability) nor the viral titer results (Tables 2 and 3) provide any statistical significance tests (e.g., p-values). Please supplement the description of the number of experimental replicates and perform comparative statistical analysis on the data to confirm whether the observed effects are statistically significant.
- Discussion of the binding constant (Kd) is missing: Fluorescence quenching data can be used to estimate the binding constant (Kd) between Orientin and the Spike protein, a critical biophysical parameter that quantitatively reflects interaction strength. It is recommended to calculate the Kd value by fitting the quenching data and correlate it with the binding free energy obtained from molecular docking.
- Line 165: “incubated at 4°C for 1” is missing a time unit and should be “1 hour”.
- Both the abstract and introduction mention that global deaths have exceeded 7 million. It is recommended that the introduction specify the date to which the data is current.
- The energy units in some tables of the manuscript are written as “kal/mol” (e.g., Table 1). These should be uniformly changed to “kcal/mol.”
- When indicating concentrations in the manuscript, “μM” is sometimes used (e.g., line 120), while ‘uM’ is used at other times (e.g., Figure 4A). This unit should be standardized throughout the text as “μ”
- There is an obvious error in the list of abbreviations on page 13: “Asn Aspartate” – ‘Asn’ is the abbreviation for asparagine, while the abbreviation for aspartate is “” This constitutes a significant error in biochemical terminology.
- The citation format is inconsistent. For example, the font used for the titles of References 48 and 49 differs from that of other references. Some references use abbreviated journal names while others use full names. These should be standardized.
- The title of Supplementary Figure 1 does not conform to the formatting used for other figure titles. It should be checked and modified to match the consistent format.
Author Response
We sincerely thank you for the thorough and constructive evaluation of our manuscript entitled “Interaction Between Orientin and Spike of SARS-CoV-2: An in silico and Experimental Approach.” Your insightful comments contributed to improving the clarity, rigor, and overall quality of our work.
We are pleased to report that all points raised have been carefully addressed. Below is a summary of the changes implemented:
Comment 1: The text contains a few instances of non-standard language usage, such as spelling errors like “temp29erature” (P9, line 319).
Response 1: All spelling inconsistencies (e.g., "temp29erature") and formatting issues have been corrected.
Comment 2: We recommend comprehensive language refinement and proofreading throughout the text to ensure consistent terminology. For example, “TCID50, EC50” should be correctly written as “TCID50, EC50” (P4, P12).
Response 2: Scientific terminology such as TCIDâ‚…â‚€ and ECâ‚…â‚€ has been standardized throughout the text.
Comment 3 and 4: Lack of statistical significance analysis: Neither Figure 5 (cell viability) nor the viral titer results (Tables 2 and 3) provide any statistical significance tests (e.g., p-values). Please supplement the description of the number of experimental replicates and perform comparative statistical analysis on the data to confirm whether the observed effects are statistically significant.
Response 3 and 4:
Statistical analyses have been added to Figure 5 (cell viability) and Tables 2 and 3 (viral titers). We included: (1) the number of experimental replicates in the Methods section, (2) the statistical tests applied, and (3) p-values indicating.
Comment 5: Discussion of the binding constant (Kd) is missing: Fluorescence quenching data can be used to estimate the binding constant (Kd) between Orientin and the Spike protein, a critical biophysical parameter that quantitatively reflects interaction strength. It is recommended to calculate the Kd value by fitting the quenching data and correlate it with the binding free energy obtained from molecular docking.
Response 5: The fluorescence quenching dataset was fitted to the Stern–Volmer model to obtain the binding constant (Kd). A discussion correlating this Kd value with the binding free energy from molecular docking was added to the Results (Lines 315 – 319) and Discussion (Lines 439 - 458) sections.
Comment 6: Line 165: “incubated at 4°C for 1” is missing a time unit and should be “1 hour”.
Response 6: Line 165 was revised to: “incubated at 4 °C for 1 hour.”
Comment 7: Both the abstract and introduction mention that global deaths have exceeded 7 million. It is recommended that the introduction specify the date to which the data is current.
Response 7: The abstract (Line 15) and introduction (Line 40) now specify the reference date for the cited global mortality data.
Comment 8: The energy units in some tables of the manuscript are written as “kal/mol” (e.g., Table 1). These should be uniformly changed to “kcal/mol.”
Response 8: All occurrences of “kal/mol” were corrected to “kcal/mol.”
Comment 9: When indicating concentrations in the manuscript, “μM” is sometimes used (e.g., line 120), while ‘uM’ is used at other times (e.g., Figure 4A). This unit should be standardized throughout the text as “μ”
Response 9: Standardization of concentration units All concentration units were revised to use the consistent form “μM”.
Comment 10: There is an obvious error in the list of abbreviations on page 13: “Asn Aspartate” – ‘Asn’ is the abbreviation for asparagine, while the abbreviation for aspartate is “” This constitutes a significant error in biochemical terminology.
Response 10: The error “Asn Aspartate” was corrected to reflect the accurate notation for asparagine (Asn) and aspartate (Asp).
Comment 11: The citation format is inconsistent. For example, the font used for the titles of References 48 and 49 differs from that of other references. Some references use abbreviated journal names while others use full names. These should be standardized.
Response 11: The entire reference list was revised to ensure conformity with Viruses guidelines, including consistent journal name formatting and uniform typography.
Comment 12: The title of Supplementary Figure 1 does not conform to the formatting used for other figure titles. It should be checked and modified to match the consistent format.
Response 12: The title and caption of Supplementary Figure 1 were reformatted for consistency with the other supplementary figures.
We sincerely appreciate the reviewer’s valuable recommendations, which contributed to improving the clarity of our manuscript.
Dr. Fabiana Avila Carneiro

Reviewer 2 Report
Comments and Suggestions for Authors
The authors' article is devoted to a computational study (using a molecular docking model using the DockThor program) of the interaction of the flavonoid Orientin with the viral protein Spike and biological screening of its effect on viral replication and cytotoxicity.
The following comments apply:
- The normal structural chemical formula of Orientin should be provided!
- For a more complete analysis of the obtained results (both computational and in vitro), a registered comparison drug with proven antiviral activity and, preferably, a similar flavonoid structure is required.

Author Response
We sincerely thank you for the thorough and constructive evaluation of our manuscript entitled “Interaction Between Orientin and Spike of SARS-CoV-2: An in silico and Experimental Approach.” Your insightful comments contributed to improving the clarity, rigor, and overall quality of our work.
We are pleased to report that all points raised have been carefully addressed. Below is a summary of the changes implemented:
Comment 1: The normal structural chemical formula of Orientin should be provided!
Response 1: The structural chemical formula of Orientin has now been included in the manuscript. This information is presented in Figure 1 (Line 70), which has been updated accordingly.
Comment 2: For a more complete analysis of the obtained results (both computational and in vitro), a registered comparison drug with proven antiviral activity and, preferably, a similar flavonoid structure is required.
Response 2: As requested, and in order to provide a more complete analysis of our results, we have incorporated three complementary reference ligands. First, we selected cepharanthine as a registered drug with proven anti–SARS-CoV-2 antiviral activity, which is now included in the revised Discussion, for qualitative comparison of antiviral activities. Second, because cepharanthine is not structurally related to flavonoids, we chose quercetin and myricetin as flavonoid reference ligands. These flavonoids share the same general scaffold as orientin and have been experimentally shown to inhibit spike-mediated viral entry, which allows a more meaningful, scaffold-consistent qualitative comparison of affinities at the spike RBD (Lines 439 – 458).
We sincerely appreciate the reviewer’s valuable recommendations, which contributed to improving the clarity of our manuscript.
